# Event-Based Landslide Modeling in the Styrian Basin, Austria: Accounting for Time-Varying Rainfall and Land Cover

**Raphael Knevels [1],\*** , **Helene Petschko [1]** , **Herwig Proske [2]** , **Philip Leopold [3]** ,
**Douglas Maraun [4] and Alexander Brenning [1]**

[1] Department of Geography, Friedrich Schiller University Jena, 07743 Jena, Germany;
helene.petschko@uni-jena.de (H.P.); alexander.brenning@uni-jena.de (A.B.)

[2] Remote Sensing and Geoinformation Department, JOANNEUM RESEARCH Forschungsgesellschaft mbH,
8010 Graz, Austria; herwig.proske@joanneum.at

[3] AIT Austrian Institute of Technology GmbH, Center for Mobility Systems, 1210 Vienna, Austria;
philip.leopold@ait.ac.at

[4] Wegener Center for Climate and Global Change, Regional Climate Research Group, University of Graz,
8010 Graz, Austria; douglas.maraun@uni-graz.at

\* Correspondence: raphael.knevels@uni-jena.de

**Abstract:** In June 2009 and September 2014, the Styrian Basin in Austria was affected by extreme events of heavy thunderstorms, triggering thousands of landslides. Since the relationship between intense rainfall, land cover/land use (LULC), and landslide occurrences is still not fully understood, our objective was to develop a model design that allows to assess landslide susceptibility specifically for past triggering events. We used generalized additive models (GAM) to link land surface, geology, meteorological, and LULC variables to observed slope failures. Accounting for the temporal variation in landslide triggering, we implemented an innovative spatio-temporal approach for landslide absence sampling. We assessed model performance using k-fold cross-validation in space and time to estimate the area under the receiver operating characteristic curve (AUROC). Furthermore, we analyzed the variable importance and its relationship to landslide occurrence. Our results showed that the models had on average acceptable to outstanding landslide discrimination capabilities (0.81–0.94 mAUROC in space and 0.72–0.95 mAUROC in time). Furthermore, meteorological and LULC variables were of great importance in explaining the landslide events (e.g., five-day rainfall 13.6–17.8% mean decrease in deviance explained), confirming their usefulness in landslide event analysis. Based on the present findings, future studies may assess the potential of this approach for developing future storylines of slope instability based on climate and LULC scenarios.

**Keywords:** extreme rainfall events; landslide analysis; rainfall-induced landslides; environmental change; generalized additive model; time-varying predictors

## 1. Introduction

In June 2009 and September 2014, more than three thousand landslides were triggered by extreme rainfall in the Styrian Basin, Austria. These landslide occurrences caused a major threat to the local population and significant damage to urban settlements, infrastructure and environment [1,2] (e.g., more than 13.4 million euros in 2009 [1]). Climate change may alter the pattern of extreme rainfall events [3] (e.g., higher frequency or intensity). Additionally, land use/land cover (LULC) change and an increased human exposure have increased human vulnerability to landslides [4–6]. With similar

extreme landslide triggering events expected in the future, landslide assessment under a changing environment is of high interest for local decision makers and stakeholders [6,7].

For local decision makers and stakeholders, the spatiotemporal delineation of potential landslide occurrences ("where" and "when"), their magnitude ("how large") and frequency ("how often") are important for landslide adaptation and mitigation strategies [4]. However, it is difficult to gather all components in a suitable landslide event inventory for "hazard" or "risk" modeling [8–10]. Therefore, most research has focused on the spatial component in landslide "susceptibility" modeling [11]. The term landslide susceptibility is commonly understood as the likelihood of a certain area to be affected by landslide occurrence on the basis of local terrain conditions [12]. These conditions are assumed to be purely spatial ("where") and time-invariant [11,13,14], following the central assumption that future slope failures are likely to occur under similar conditions as past landslides [15,16].

However, several authors have shown that the concept of time invariance is often violated on the time scale of few to several decades, e.g., Reichenbach et al. [17] discovered a changing landslide susceptibility through anthropogenic land use changes in the period 1954–2009, while Samia et al. [18] demonstrated a higher susceptibility in previously unstable areas for a period of about ten years. On decadal to centennial time scales, natural and anthropogenic climatic changes may influence meteorological variables which further affect landslide susceptibility [11]. Therefore, we follow the recommendation of Gariano and Guzzetti [19] to construct new models considering and investigating changes of environmental variables for landslide susceptibility [19]. In our approach, local terrain conditions (e.g., predisposing factors such as slope angle, slope aspect, . . . ), which are assumed not to change substantially in the course of centuries, are still considered time-invariant, but are extended by time-varying predictor variables (e.g., preparatory and triggering factors such as precipitation or LULC).

There are several approaches to include rainfall in landslide analysis: most studies are based on physically based models [20,21] or on rainfall-threshold extraction [22,23], respectively, with the aim of creating a reliable forecasting system (refer to [24] for review). However, while physically based models are applicable only at a local scale [25], the rainfall-threshold extraction is often conducted without the consideration of other important predisposing factors or the spatial variability of rainfall. Therefore, recently, some authors combined landslide susceptibility maps with rainfall thresholds to enhance forecasting effectiveness also on a regional scale [26–28]. Reichenbach et al. [11] reviewed methods to create landslide susceptibility maps. Focusing on statistically based methods, the application of event-based landslide susceptibility models that consider time-varying predictor variables provides the possibility (i) to extract the underlying relationship while also accounting for other predisposing factors, and (ii) to assess environmental change scenarios.

So far, the application of time-varying predictor variables in event-based landslide susceptibility modeling is still an active research field. While several authors already investigated landslide susceptibility under the influence of LULC change [17,29–33], only few have analyzed the links between meteorological variables and slope failure [34,35] or their combined role in explaining landslides [6,36]. With meteorological variables or in combination with LULC, Gassner et al. [34] and Shou & Yang [35] used logistic regression to link meteorological variables such as rainfall amount (e.g., sums in mm per day [34]) or the landslide-rainfall index [35] to landslides. Kim et al. [36] used a Maximum Entropy model with a set of meteorological and LULC variables for landslide modeling and variable importance assessment. Promper et al. [6] adapted the model of [34] for a landslide exposure analysis using climate and LULC scenarios. These studies found that, first, meteorological and/or LULC variables were important in explaining landslides [36], and second, future landslide susceptibility will likely increase due to shorter return periods of intense rainfall [34,35] and an increased exposure of human infrastructure and farmlands [6,36]. Moreover, the above-mentioned modeling approaches achieved good performances, justifying their application for climate and LULC change scenarios [6,31,34–36].

Our objective was the investigation of landslide occurrences caused by the extreme rainfall events of 2009 and 2014 by establishing a suitable model and sampling design using a geostatistical approach. The main focus was on the relationship of time-varying meteorological and LULC variables to slope failures. As in the pre-Alpine Styrian region the contribution of these variables to landslide activity is still quantitatively unknown [37], we furthermore aimed to improve the basic understanding of landslide susceptibility in that region. Given the expected non-linear relationships, we chose the generalized additive model (GAM) for our study. We analyzed differences in model performances applying spatial and spatiotemporal cross-validations with different sets of input variables, and evaluated the effect of time-varying predictor variables on landslide occurrences using variable importance measures, odds ratios (OR), and component smooth function (CSF) plots.

## 2. Materials and Methods

### 2.1. Study Area and Extreme Rainfall Events

The study area is located in the Styrian Basin, Austria, and covers around 3831 km$^2$ extending from 15°9′ E 47°10′ N to 16°9′ E 46°36′ N (Figure 1A,B). It is characterized by a hilly topography and flat lowlands of fertile arable land, and a West-East gradient in elevation ranging from 1167 m to 196 m (relative relief [38] of 3–550 m/km$^2$), bordering the Styrian Prealps from southwest (Possruck Mountains, Koralpe) to northwest (Stubalpe Mountains, Graz Mountains). The underlying geology of Neogene sediments (thick Miocene and minor Pliocene sediments) mainly consisting of a heterogeneous mixture of sands, silts, clays, marl, and gravels, makes the Styrian Basin especially prone to landslides [1,34]. For a detailed geological overview, refer to Gasser et al. [39].

The study area was affected by extreme rainfall events in June 2009 and September 2014. The rainfall event in summer 2009 occurred from 22 to 25 June, followed by several less severe events (Figure A1a) [1,40]. In June 2009, a persistent cut-off low over central Italy remained stationary over the central Adriatic and Balkans for several days. During that time period, warm and moist air from the Mediterranean and Eastern Europe advected anti-clockwise into Central Europe. In the Eastern Alps, upper tropospheric divergence in the cut-off low and orographic lifting at the northern edge of the Alps coincided, resulting in a series of heavy thunderstorms and rainfalls in the Styrian Basin [1,40]. Within 24 h, some locations recorded more than 100 mm of rainfall, which corresponded to a 50-year return period [40]. In Styria, around 1700 landslide-related private damage claims were filed. A focus region with more than 3000 recorded landslides was the northern part of the district of South East Styria (Figure 1B), where the state of emergency was declared for several municipalities [1].

In September 2014, a similar, but less severe rainfall event occurred. From the first to 21 September, the Styrian Basin was regularly affected by heavy thunderstorms [2]. Especially in the period from 12 to 15 September (Figure A1b), an upper-level low brought multiple episodes of heavy rainfall into the region (total rainfall sums of 30–100 mm) [41,42], causing damage by hail, floods, and landslides [2]. Landslide-related damages were distributed across the entire Styrian Basin, but there was a discernible cluster of landslide occurrences with more than 500 landslides south of Leibnitz and north of Gleisdorf (Figure 1B).

### 2.2. Data

For our analysis, we used climate, land surface and landslide data from various sources and at different spatial resolution (Table A1 for overview). We decided to use 10 m × 10 m resolution as our target resolution and applied bilinear interpolation for resampling. This downscaling was necessary due to the dependence of landslide susceptibility on local-scale topography, although we acknowledge that we are unable to capture local-scale patterns of climate or geology.

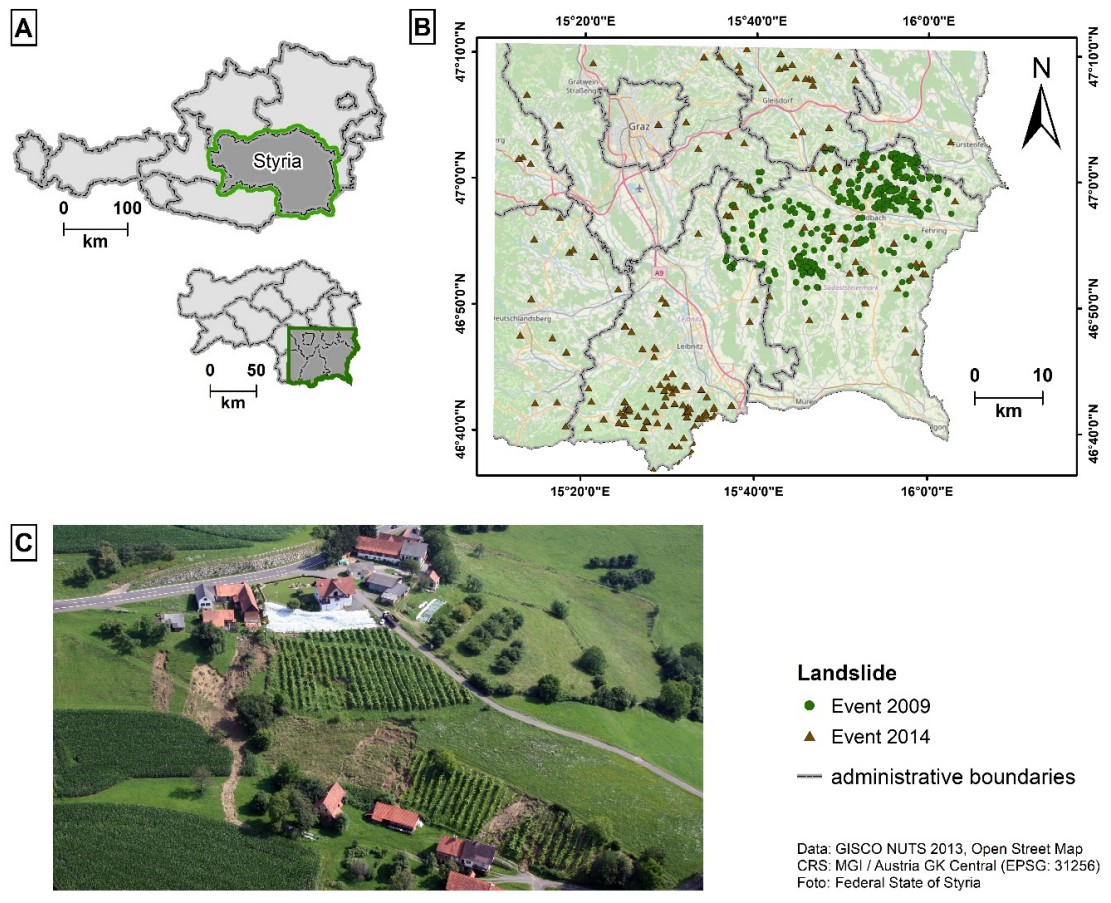

**Figure 1.** Overview of the study area and rainfall-induced landslides: (**A**) Location in the federal state of Styria (lower part) and Austria (upper part); (**B**) Study area with landslide occurrences during extreme rainfall events; (**C**) Examples of shallow landslides in 2009.

### 2.2.1. Climate Data

As climate data we used INCA (Integrated Nowcasting through Comprehensive Analysis) precipitation data provided by the Central Institute for Meteorology and Geodynamics (ZAMG, German: Zentralanstalt für Meteorologie und Geodynamik) from Austria. INCA is a nowcasting system, especially suitable for mountainous regions [43]. The precipitation data is derived combining surface stations with remote sensing data in 15-min intervals on a 1 km × 1 km grid. For details, refer to Haiden et al. [43]. For our analysis, we aggregated precipitation to hourly intervals.

### 2.2.2. Land Surface Data

An airborne LiDAR-derived high-resolution digital terrain model (HRDTM) with a 1 m × 1 m resolution was provided by the GIS department of Styria (GIS-Steiermark). It was based on surveys made in the years 2009–2011 (2009: 15–21 March, 2010: 09 April to 08 July, and 2011: 12 April to 19 August). The 2009 survey covered 66% of the study area (2010: 25%, 2011: 10%).

The LULC data was derived from the original airborne LiDAR point-cloud by the JOANNEUM RESEARCH (JR), and focused on forest cover, specifically the conifer fraction. We reclassified the forest cover into four classes: (i) no forest, (ii) broadleaf forest (0–25% conifers), (iii) mixed forest (25–75% conifers), and (iv) conifer forest (75–100% conifers).

A watershed map and a geological basemap at the reference scale 1:50,000 (compiled from maps illustrating the Natural Environments Potential of Styria [44,45] and the Geological Map of Styria 1:200,000 [46]) was also provided by GIS-Steiermark. In the geological basemap, alluvial deposits were

corrected in order to match valley floors visible in the HRDTM, and we reclassified the geological units into five relevant classes: (i) Neogene formations dominated by fine-grained sediments, (ii) Neogene formations with coarse-grained layers, (iii) pre-Würmian Pleistocene formations, (iv) Würmian and Holocene sediments, and (v) other units.

We downloaded OpenStreetMap (OSM) data extracts for Styria provided by the Geofabrik as of 29 October 2018. We used a combination of areal (land use, buildings) and linear (roads, railways) data in the sampling design (see Section 2.3.1). In order to account for the approximate spatial extent of real-world objects that were represented as polyline features, we buffered the line data as follows: For roads, we applied a 1.5 m buffer for the classes 'unclassified,' 'living street,' and 'residential,' a 2.5 m buffer for 'trunk,' 'secondary,' 'tertiary,' a 3 m buffer for lines of class 'primary,' and a 5 m buffer for 'motorway.' Railways were buffered with 5 m.

### 2.2.3. Landslide Data

The landslide data of the extreme rainfall events in 2009 and 2014 were provided by different federal institutes (Table A2). After the 2009 rainfall event, the Institute of Military Geoinformation (IMG) and the Geological Survey of Austria (GBA) were involved in recording landslides. The IMG was ordered to record landslides based on reported damages [47] (photo documentation, geographical localization, failure date). For this, the IMG created a mobile GIS (not available to us) using geocoded addresses based on the damage reports, and subsequently inspected the landslide locations in the field [47]. Additional landslides recognized on the way were also recorded. The IMG mapped 301 medium and large landslides as polygons ($> 25$ m$^2$), 148 small ones as points, and selectively added 577 scarp lines [47]. Based on the IMG landslide records, the GBA identified two focus regions for a follow-up investigation [48]. In this investigation, the GBA identified 680 landslides of the Cruden and Varnes [49] slide type, partly overlapping the IMG inventory [48]. The GBA landslides were inventoried using points geocoded in the landslide scarp area [48].

We analyzed and filtered these landslide data regarding their relevance (scarp length $> 10$ m, at least 1 m displacement), type (shallow or deep-seated), and reliability (location and process matched available photo documentation and attributes of database). In the case of the IMG landslide data, 743 records from a total of 1026 were selected (148 points, 577 scarp lines, and 301 polygons), and in the case of the GBA landslide data 342 out of 680 records were classified as "relevant and reliable." For our study, we subsequently finalized the landslide data by removing duplicates and unified the geometries of the IMG and GBA records. We decided to use one point per landslide, and thus transformed the IMG data accordingly (median points on scarp lines, and 'point on surface' [50] in the upper third of polygons). If multiple geometry types represented the same landslide, we prioritized as follows (in decreasing order): IMG point, IMG scarp line as point, IMG polygon as point, GBA point.

The final 2009 dataset included 487 landslides (45% GBA, 55% IMG). The majority of these were shallow movements (85%), located in non-forested areas (86%), and for 22% of the landslides the failure date was known (Table A2). Missing dates were randomly sampled from the known failure dates and assigned with the constraint of reproducing the distribution of the known failure dates (Table A3).

After the 2014 rainfall event, the Department of Hydrology, Resources and Sustainability of the Styrian Government (LS) studied only landslides that damaged infrastructure, agricultural land or buildings, resulting in claims that were classified as worthy of remediation. Based on the damage reports, we found it challenging to determine a landslide's exact geographic location and its characteristics (e.g., shallow vs. deep-seated). Out of a total of 508 events reported to the authorities, only 139 landslides were selected for our analysis (minimum size, expert report, removal of duplicates). Using the reported damage description, orthophotos and OSM data, we categorized the affected locations into vineyard and orchard (30.8%), farmland (22.2%), urban areas (streets and buildings, 30.3%) and meadow and forests (16.8%). The majority of landslides were deep-seated (60%, 5% shallow, 35% no-data), and for each landslide, the failure date was available (Table A2).

*2.3. Landslide Susceptibility Modeling*

For landslide susceptibility modeling, we used the semi-parametric generalized additive model (GAM) [51]. A GAM is an extension of the generalized linear model (GLM) since it is capable of replacing the linear terms with non-linear transformations, the so-called component smooth functions (CSF). The CSF are automatically fitted in order to uncover non-linear relationships of the response on the predictors [52], so there is no manual "detection work" necessary to find the underlying relationship [51]. Furthermore, the additive structure of a GAM allows the assessment of predictor-response relationships while accounting for the remaining predictors. The application of GAMs in landslide susceptibility modeling has already demonstrated its benefits in terms of performance [53,54] and model interpretability [14,55].

Our landslide modeling approach was conducted in the free and open source computing environment R [56] (R version 3.5.2). As the modeling framework we used the *mlr* package [57] and the GAM implementation of the *mgcv* package [52]. We used GRASS (Geographic Resources Analysis Support System) GIS 7.4.1 [58], SAGA (System for Automated Geoscientific Analysis) GIS 6.3.0 [59], and TauDEM 5.3 (Terrain Analysis Using Digital Elevation Models) [60] for geomorphometric operations. While the former GIS software are executable from within R by the *rgrass7* [61], and *RSAGA* [62] packages, respectively, TauDEM 5.3 geocomputing tools were accomplished through operating system commands in R.

In the following Section 2.3.1 our sampling framework is shown. Subsequently, the landslide models and their input data are presented (Section 2.3.2). The estimation of model performances and variable importance is explained in Section 2.3.3.

2.3.1. Sampling Design

For training GAMs, observations of landslide presence and absence are required. While presence observations were prepared from the local authorities' landslide data (see Section 2.2.3), reliable absence observations also need to be chosen. However, often little attention is paid to the equally important task of absence positioning [63,64].

The rainfall-induced landslides in 2009 and 2014 occurred on several days and throughout the study area. Additionally, since the rainfall was heterogeneously distributed in space and time, each landslide location was affected differently. Therefore, the spatial and temporal distribution must be considered for absence positioning. The challenges were, first, the creation of a suitable mask of landslide-free candidate areas for absence observations ('landslide-free mask'), and second, the determination of the corresponding "failure dates." Different strategies were applied for the 2009 and 2014 rainfall-triggered landslide events, as outlined in the following paragraphs.

To obtain a landslide-free mask for the 2009 event, different strategies were applied for both data sources (Figure 2A). For the GBA landslide data, we assumed the focus areas to be completely mapped. Thus, we defined the area outside a buffer of 140 m around each landslide point as stable. As an outer boundary, we generated a convex hull around the landslide locations.

For the IMG landslide data, we were only certain about the investigated locations and therefore believed that additional unreported landslides exist in the surrounding areas. Since the IMG prepared a mobile GIS (not available to us) for their field work, we assumed that all landslides visible along the driven route were recorded. To delineate the inventoried area, we therefore performed a viewshed analysis along a hypothetically driven route. Such viewshed analysis ("effective surveyed area") has already been applied for landslide absence positioning, especially when the terrain was inaccessible for field investigations [64,65]. We extracted a hypothetically driven road network under the assumptions that the shortest route was taken and that each landslide was visited only once. For this, we calculated a pairwise driven-distance matrix using an open source routing machine (OSRM) [66], and found the shortest route by solving the traveling salesman problem (TSP) with the nearest neighbor method [67] using the R *TSP* package [68]. For the final road network, we "re-drove" the shortest route using OSRM allowing for two alternative ways at each landslide location. Based on the final routing result,

the "effective surveyed area" was extracted using the approach of Bornaetxea et al. [64]. We transferred the original GRASS GIS based python tool *r.survey.py* [64] into R to keep consistency within our modeling framework (see *survey* in R *RainSlide* package [69]). Following [64], we placed viewpoints every 200 m along the final road network, and calculated viewsheds allowing for a 1000 m viewing distance. For the final landslide-free mask, we merged the GBA landslide-free mask (i.e., its convex hulls) with the "effective surveyed area," and erased so-called trivial areas [63]—areas considered as not susceptible for landslides (e.g., floodplains, flat areas), and anthropogenic structures with similar geomorphometric characteristics as landslides (e.g., quarries) (Figure 2B). Finally, we randomly distributed absence points within the landslide-free area using a 5:1 absence-presence ratio (in total, 2435 absence points and 487 landslides). We set an arbitrarily chosen minimum distance constraints of 140 m between absence samples to account for spatial autocorrelation, and to landslide points to avoid absence observations located in potentially slid areas. We chose the ratio of 5:1 to account for random variability in the sampling design later in landslide modeling.

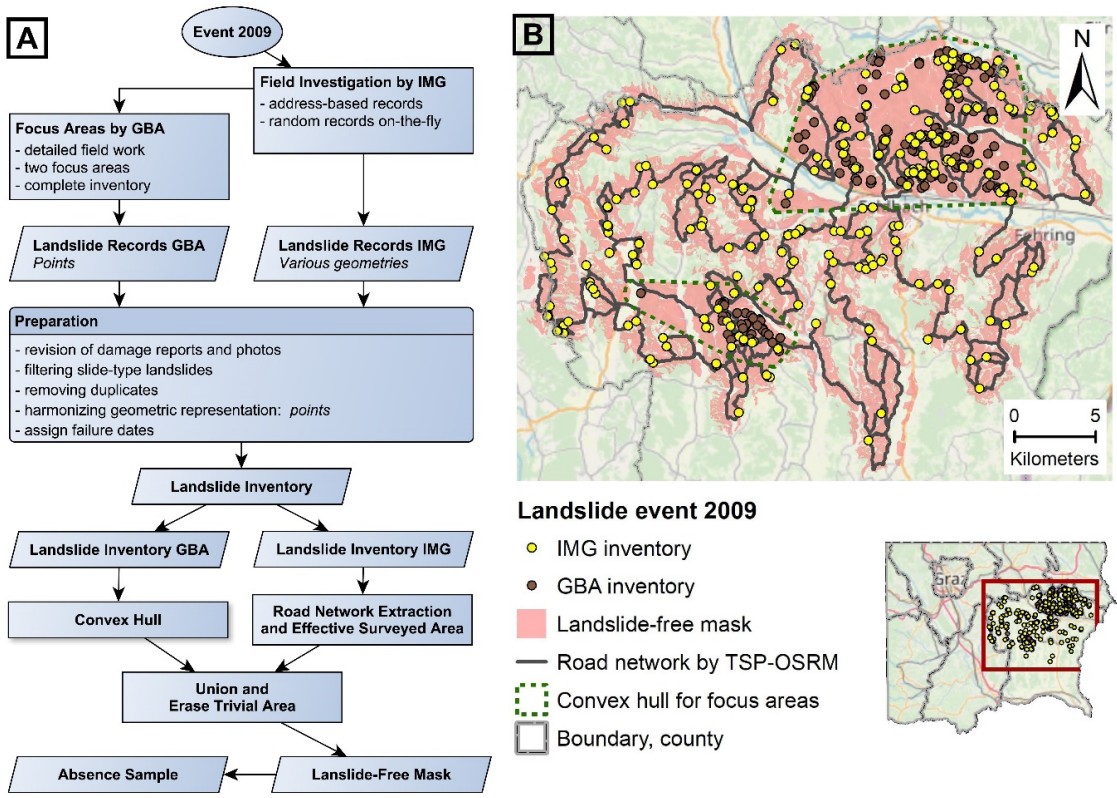

**Figure 2.** Overview of the sampling design for the 2009 rainfall-triggered landslide event: (**A**) Overview of the workflow; (**B**) Overview map with OSM data as background.

To create a landslide-free mask for the 2014 event, we used the fact that the LS records solely based on reported damages classified as worthy of remediation (Figure 3A). We assumed that wherever infrastructure or cultivated areas were harmed, a damage was likely reported. Based on OpenStreetMap (OSM) data, we created an 'infrastructure mask' (buildings, roads, railways; buffered according to type, Section 2.2.2) and 'agricultural mask' (vineyard, orchard) determining the potentially 'affected area mask.' We did not include the LULC classes of forest, meadow and farmland to avoid the introduction of an under-reporting bias due to unreported landslides in forest, or already "vanished" landslides on meadows and farmland. In the next step, we buffered the 'affected area mask' with the median slope length of the inventoried landslides to account for the downslope mass movement. Since we did not want to consider infrastructure in the landslide-free area, we erased the 'infrastructure mask' area from the 'affected area mask' in the following step. Similar to the 2009 sampling design, trivial areas

were not considered for the landslide-free mask. Finally, we further constrained the landslide-free mask based on a 'watershed mask' (landslide-affected watersheds and their neighbors) to account for the general distribution of landslides in the Styrian Basin (Figure 3B). To obtain an absence sample, we performed stratified random sampling with the same settings as for 2009, but using the 'watershed mask' for stratification (in total, 695 absence points and 139 landslides, Table A2).

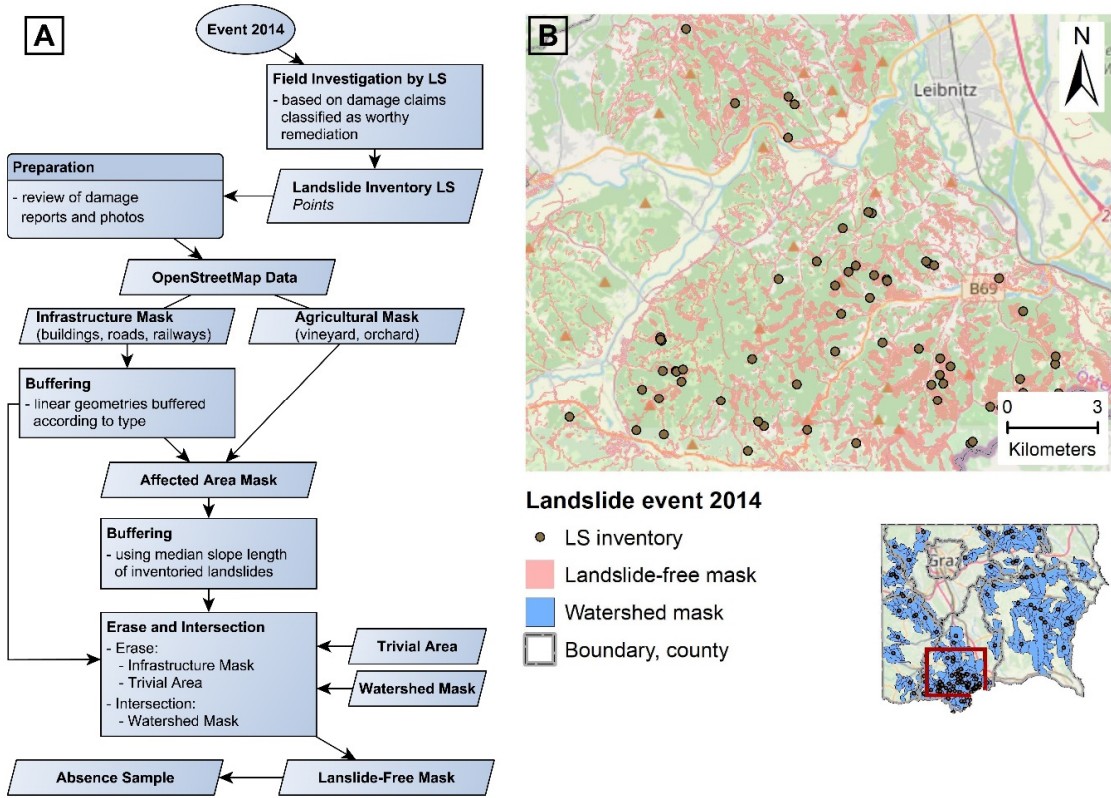

**Figure 3.** Overview of the sampling design for the 2014 rainfall-triggered landslide event: (**A**) Overview of the workflow; (**B**) Overview map with OSM data as background.

For the temporal sampling of the "failure dates," we analyzed the rainfall event duration for each landslide following the automatic procedure of Melillo et al. [22]. We defined the temporal extent of the "failure dates" from the beginning of the rainfall event until the last day of landslide occurrence. Using this approach, for 2009 the temporal extent was from 19 to 26 June. For 2014, the temporal extent differed according to the stratum of the 'watershed mask.' It ranged from 30 August to 15 September in the most extreme case. Subsequently, the dates of the temporal extent were randomly assigned to the absence observations (Table A3).

2.3.2. Landslide Models and Input Data

For landslide modeling, meaningful predisposing factors serving as predictor variables have been reported [70] and have already been successfully applied [14,53,54]. In our study, we linked suitable land surface variables (LSV), meteorological variables, geology, and LULC to slope failures (Table 1).

**Table 1.** Predictor variables used in this study.

| Variable(s) | Software | Setting | Method |
|---|---|---|---|
| *land surface variables* | | | |
| convergence index (100 m, 500 m) | SAGA GIS | r = 100 m, 500 m | [71] |
| curvature (plan, profile) | SAGA GIS | | [72] |
| flow accumulation, D-Infinity | TauDEM | log-transformed | [73] |
| normalized height | SAGA GIS | w = 5; t = 2; e = 2 | [74] |
| slope angle | SAGA GIS | | [72] |
| slope angle, catchment area | SAGA GIS | | [75] |
| slope aspect (S-N, W-E) | SAGA GIS | cosine, sine transformed | [72,76] |
| topographic position index (TPI) | SAGA GIS | r = 500 m | [77] |
| topographic wetness index (SWI) | SAGA GIS | | [75] |
| meteorological variables | | | |
| five-day rainfall | R | | |
| maximum three-hour rainfall intensity | R | | |
| *geology* | | | |
| geology | | ref: 'Neogene formations with coarse-grained layers' | |
| *land use/land cover* | | | |
| forest type | | ref: 'no forest' | |

Setting, scale-dependent parameters: r: radius; w,t,e: Parameters in SAGA GIS module Relative Heights and Slope Positions; ref = reference level (see Section 2.2.2).

The LSV were derived from the resampled HRDTM. The geology consisted of five units (see Section 2.2.2), of which we used the unit 'Neogene formations with coarse-grained layers' as reference level. The LULC variable forest type consisted of four classes (see Section 2.2.2) with 'no forest' as reference level. However, for the 2014 event data, we only distinguished between 'forest' and 'no forest' due to low observations in the different forest classes. The calculation of the meteorological variables was based on the INCA precipitation data. Here, we extracted two variables (Figure A2): (1) The five-day precipitation sum up to the failure dates, providing a proxy for soil moisture, and (2) the maximum three-hour rainfall intensity at the failure date, representing a potential trigger. Other authors used similar interval sizes [34,36], and different aggregation schemes would result in correlated rainfall variable (e.g., [22]: correlation of 0.95 between five-day rainfall and total amount during rainfall event, and of 0.58 between maximum three-hour rainfall intensity and amounts of rainfall sub-events).

Our model design aimed to explore the performance of event-based models and to improve the understanding of the influence of time-varying variables (meteorological and LULC variables) on landslide occurrence. For this purpose, we created landslide models with different data sets and input variables: For the assessment of the event modeling ('event-based models'), we built (i) a GAM based on the 2009 event data (GAM-09), (ii) a GAM based on the 2014 event data (GAM-14), and (iii) a GAM based on the combined event data (GAM-Co); these models used all available predictor variables. For the assessment of the effect of different sets of input variables, we used the combined event data and compared models that included, in addition to LSV and geology, only (iv) LULC variables (GAM-LULC), and (v) meteorological variables (GAM-Meteo), respectively. The benchmark model that excluded these time-varying variables is referred to as (vi) 'GAM-Base' ('comparison models').

### 2.3.3. Model Performance and Variable Assessment

For the performance assessment, we applied different strategies depending on the model design. The 'event-based models' were studied using a cross-validation in space (SpCV) and time (SpTempCV), while for the 'comparison models' only a spatial cross-validation was applied (SpCV-Comparison).

In the first step, we resampled the available observations $n = 100$ times with a 1:1 landslide presence-absence ratio following [14]. We partitioned each subsample spatially using *k*-means clustering of the coordinates ($k = 5$), resulting in $k$ disjoint subsets or 'folds'; this was repeated $rep = 5$ times. In SpCV, in each repetition we combined four folds into a training set and used the remaining fold for validation, iterating this procedure until each fold was used once for validation (i.e., $k$ models

per repetition). For the SpCV evaluation of the 'comparison models,' we used identical training and validation data to ensure that performances are directly comparable. In SpTempCV, we spatially resampled the data of one event (e.g., 2009) to obtain training data as described above for SpCV, but used all unseen data of the other event (e.g., event 2014) for validation. We decided to apply SpCV for training to account for potential autocorrelation effects through temporal path dependency of new landslides on pre-existing ones ("follow-up landslides" [18]). Furthermore, using SpCV and SpTempCV, we obtained independent performance estimates for the validation data, and thus nearly unbiased averaged performance results [78].

As our performance measure we calculated the area under the receiver operating characteristic curve (AUROC), a well-established quality measure for binary response data [54,55,79]. The AUROC value may range between 0.5 for no discrimination and 1.0 for perfect discrimination [80]. For the interpretation of the AUROC values, we followed Hosmer et al.'s [81] interpretation guideline.

We studied the importance of each variable in explaining landslides by using the mean decrease in deviance explained (mDD, %). The mDD shows the relative contribution of a variable to the model's overall deviance explained [82]. We calculated mDD by measuring the decrease in deviance explained of a model while leaving the variable of interest out (we fixed the smoothing parameters through the *mgcv::gam sp* argument), and averaged over all SpCV models. A higher mDD indicates a greater contribution of a variable.

We furthermore visualized the relationship between landslide occurrence and time-varying variables by means of CSF plots and odds ratios (OR) considering all SpCV models. The CSF plots use the additive structure of a GAM to visualize the predictor-response relationship. In our analysis, we were especially interested in the relationship between landslide occurrences and the meteorological variables. The OR represents the chance of an outcome given a particular exposure [83]. An exposure with OR < 1 is associated with lower odds of outcome and OR > 1 with higher odds of outcome, respectively; OR = 1 means no influence [83]. We extracted ORs for the LULC variable forest type, and averaged over all SpCV models.

For the assessment of differences in AUROC estimates of the 'comparison models' and between the ORs of the forest types (i.e., log-transformed OR), we conducted Wilcoxon signed rank tests ($\alpha = 0.05$; R *coin* package [84]) following the recommendation of Demšar [85]. For the adjustment of the *p*-values for multiple comparisons, we followed Benjamini & Hochberg [86], and estimated the effect size *r* from the *p* level according to Rosenthal [87].

## 3. Results

### 3.1. Performance Assessment

The SpCV performances showed excellent to outstanding discrimination capabilities of all landslide models (Figure 4, Table A5). The 'event-based models' had median AUROC (mAUROC) values greater than 0.9 with 'GAM-Co' as best performing model (0.94, lowest interquartile range (IQR); GAM-09: 0.94; GAM-14: 0.92). However, while 'GAM-09' and 'GAM-Co' had a low variability in AUROC values (range: GAM-09: 0.14; GAM-Co: 0.15), the AUROC values of 'GAM-14' ranged from 0.44 to 1.

The SpTempCV performances showed large differences between the 'event-based models.' While the mAUROC value of 'GAM-09' (0.89 in predicting event 2014) and 'GAM-Co' (predicting 2009: 0.95; predicting 2014: 0.91) were excellent to outstanding - with 'GAM-Co' as best performing model, the landslide discrimination of 'GAM-14' was only acceptable (mAUROC 0.72 in predicting event 2009). Similar to the SpCV assessment, 'GAM-14' showed the highest variability in AUROC values (range of 0.66; GAM-09: 0.08; GAM-Co: 0.08 for event 2009 and 0.15 for 2014).

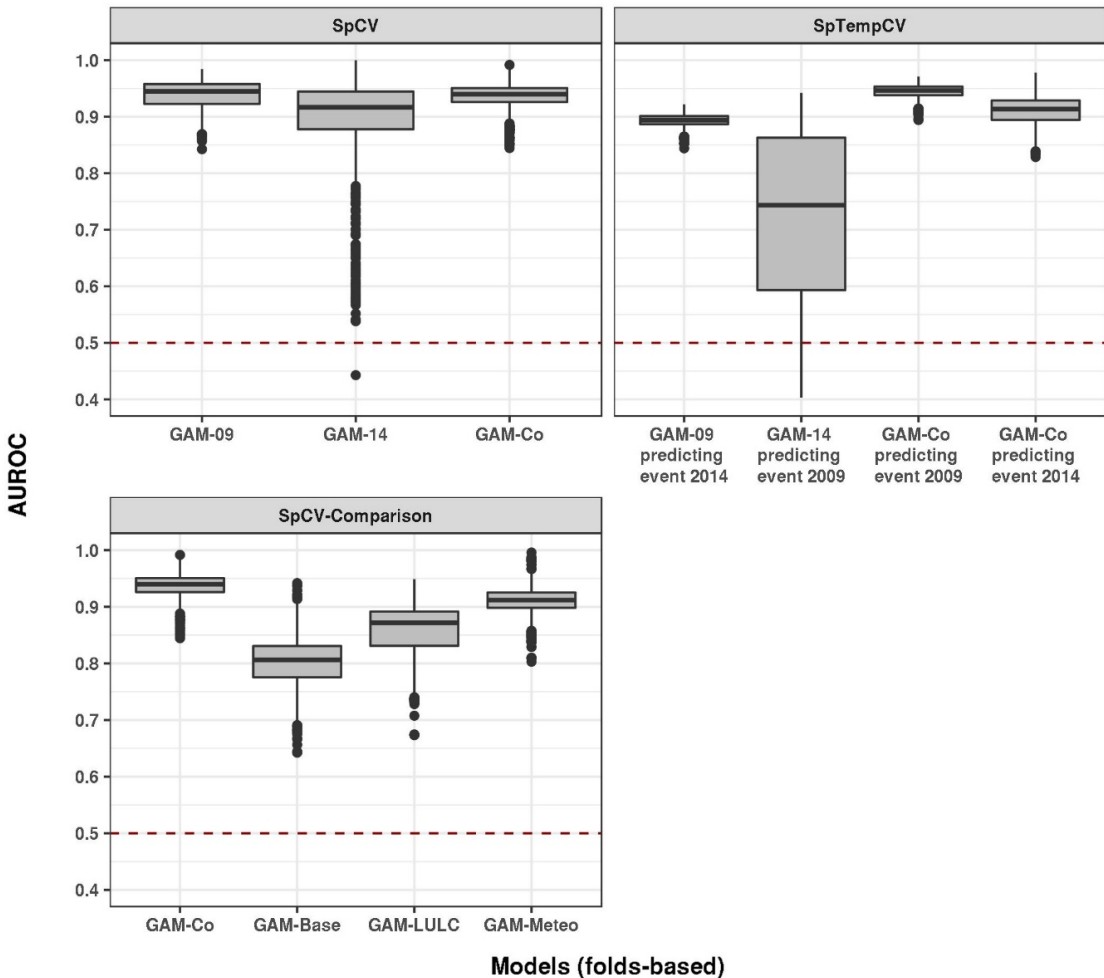

**Figure 4.** Model performance results. Boxplots show folds-based AUROC values using the validation data in the SpCV, SpTempCV and SpCV-Comparison model framework; Red dashed line: no discrimination.

In the SpCV-Comparison, 'GAM-Co' - the model with the entire set of input variables - had the highest mAUROC (0.94), and was followed in decreasing order by 'GAM-Meteo' (0.91), 'GAM-LULC' (0.87) and 'GAM-Base' (0.81). Even if all models showed excellent to outstanding discrimination capabilities, there was a significant decrease in mAUROC performances from 'GAM-Co' to 'GAM-Base' with $p < 0.001$ and a large effect $r \geq 0.82$ (Table A6).

### 3.2. Variable Importance

The variable importance assessment revealed that in each model that included meteorological variables, the five-day rainfall was the most important variable (mDD between 13.6% and 17.7%; Figure 5, Table A7). The maximum three-hour rainfall intensity ranked among the top five important variables, but with lower importance (mDD 1.0–2.9%). The LULC variable forest type was also important, ranking second in 'GAM-09,' 'GAM-14,' and 'GAM-Co,' and first in 'GAM-LULC' (mDD 4.1–12.5%). Among the LSV, slope angle was the most important variable in all models (mDD 3.9–7.9%), except for 'GAM-14,' where the TPI was more important (mDD 2.4%).

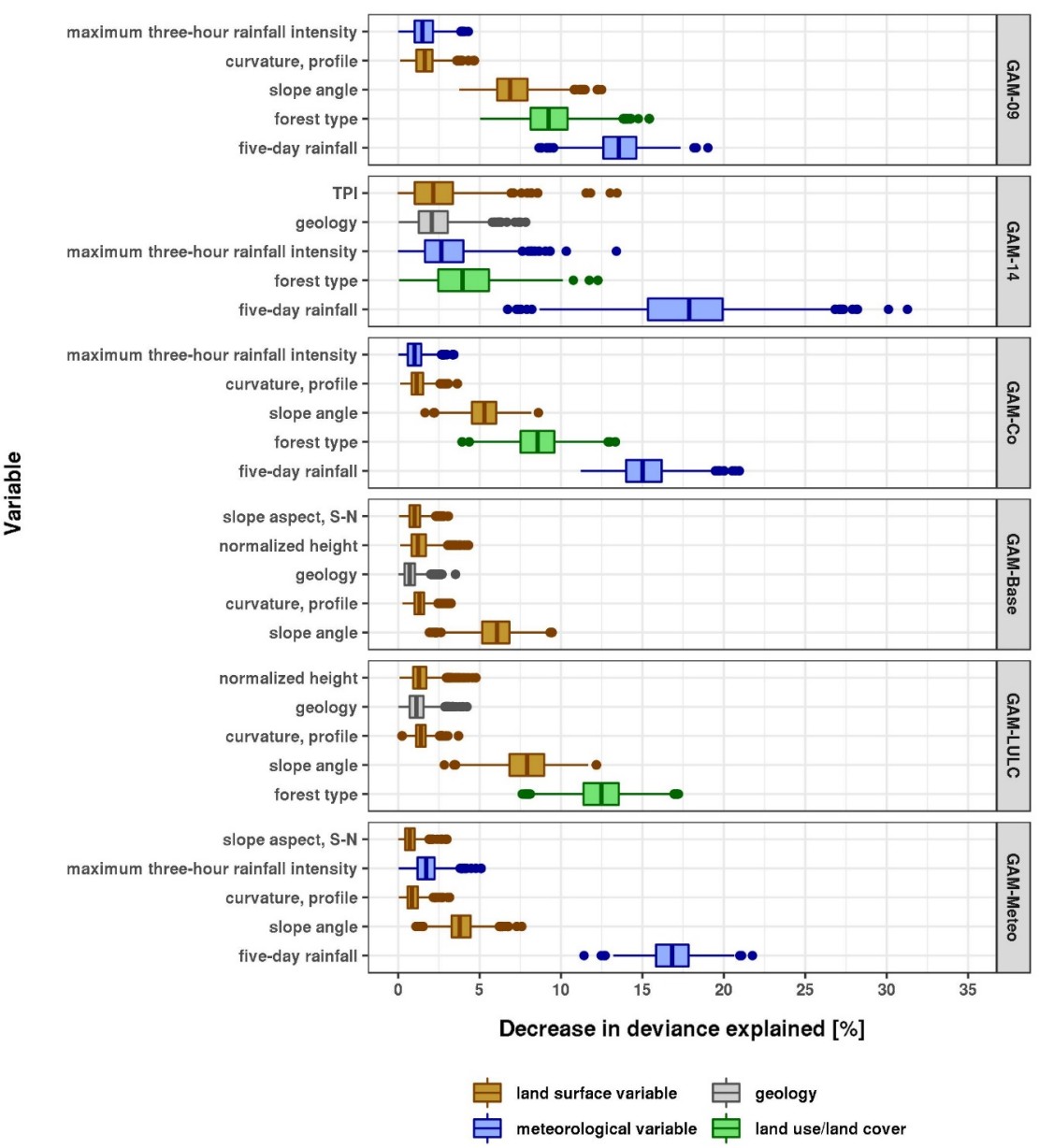

**Figure 5.** Variable importance using decrease in deviance explained [%]. Selection of top five most important variables sorted by mDD. For an overview of all input variables, refer to Table A7.

### 3.3. Relationship between Landslide Occurrence and Time-Varying Variables

#### 3.3.1. Relationship to Meteorological Variables

The relationships displayed in the CSF plots of the meteorological variables for 'GAM-09,' 'GAM-Co' and 'GAM-Meteo' were consistent, while the 'GAM-14' CSFs showed unstable behavior especially for larger values of the predictor variables (Figure 6).

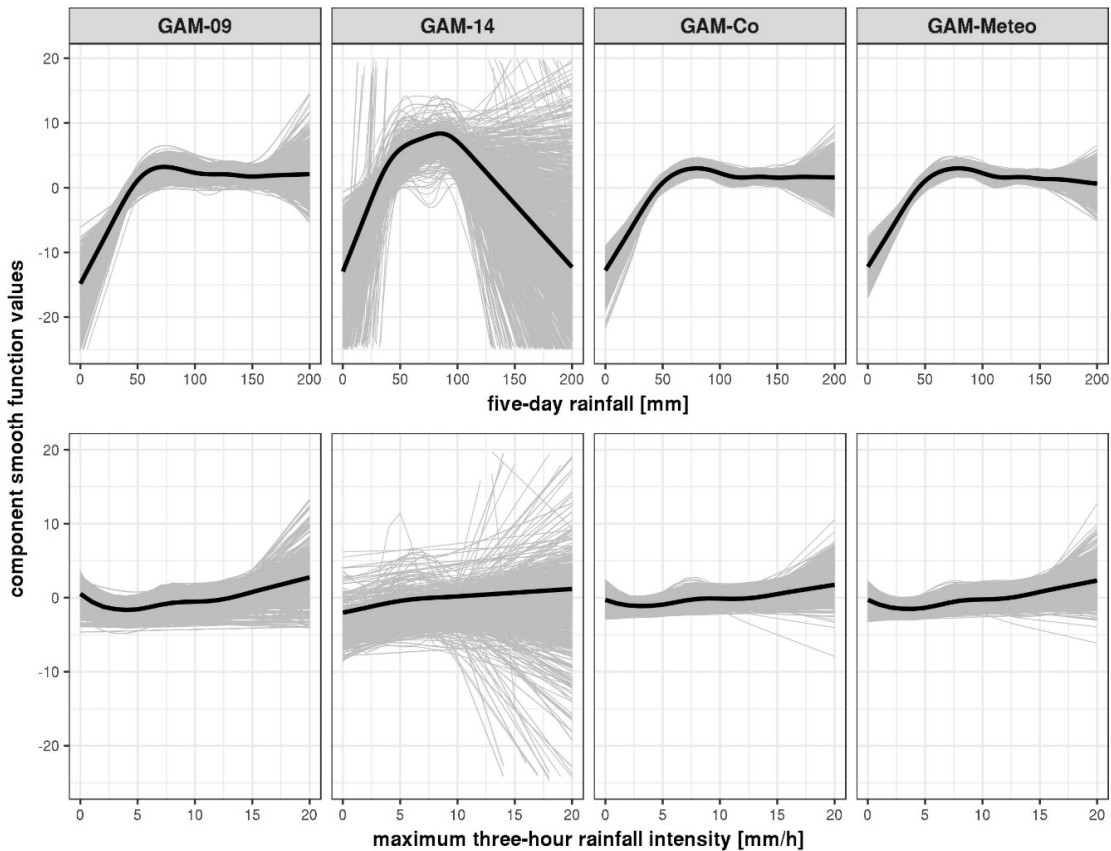

**Figure 6.** Relationship between landslide occurrence and metrological variables visualized with CSF plots. The relationships on the linear predictor scale (logit scale, centered) were extracted from all SpCV models (grey lines) and averaged (black line).

For the five-day rainfall, the majority of models showed a higher chance for landslide occurrence with a higher amount of accumulated rain up to around 75 mm (e.g., around 7.9 times higher than for 50 mm), where a maximum was reached. Afterwards, the chances of landslide occurrence slightly decreased to a plateau (level of 50 mm), and remained on a nearly constant level, but with higher variability above 150 mm. The CSFs of 'GAM-14,' in contrast, showed a decrease for higher rainfall amounts (e.g., OR = 0.01 for 125 mm vs. 100 mm). We estimated the inflection point of the relationship graphs at around 50 mm, defining a critical rainfall threshold for landslide occurrences of five-day rainfall (Figure A3).

For the maximum three-hour rainfall intensity, the majority of models indicated a lower chance of landslide occurrence with a higher rainfall intensity up to about 4 mm/h (OR 0.29 relative to no rainfall). Beyond this point, the chances of landslide occurrence increased again (e.g., around 3.1 times as high for 10 mm/h as for 4 mm/h). For the 'GAM-14,' the chance of landslide occurrence was generally higher with higher rainfall intensity following a nearly linear trend (around 1.17 times higher per 1 mm/h higher intensity). Thus, a critical rainfall threshold was not identifiable. All models showed a higher variability from around 10 mm/h.

The visualizations of the relationship indicated an overall higher variability in CSF values for the maximum three-hour rainfall intensity and, in particular, for 'GAM-14.'

### 3.3.2. Relationship to the LULC Variable

On average, forest areas had a substantially lower chance of landslide occurrence than non-forest areas (e.g., GAM-14, OR = 0.03) (Figure 7). However, we discovered significant differences between the forest types in 'GAM-09,' 'GAM-Co' and 'GAM-LULC' (Table A8): The chance of landslide occurrence

in broadleaf or mixed forest was significantly less than in conifer forest ($p < 0.001$ with a large effect size of $r \leq -0.83$). The differences between broadleaf and mixed forest were only marginal (e.g., OR of GAM-Co: mixed forest = 0.027, broadleaf = 0.032).

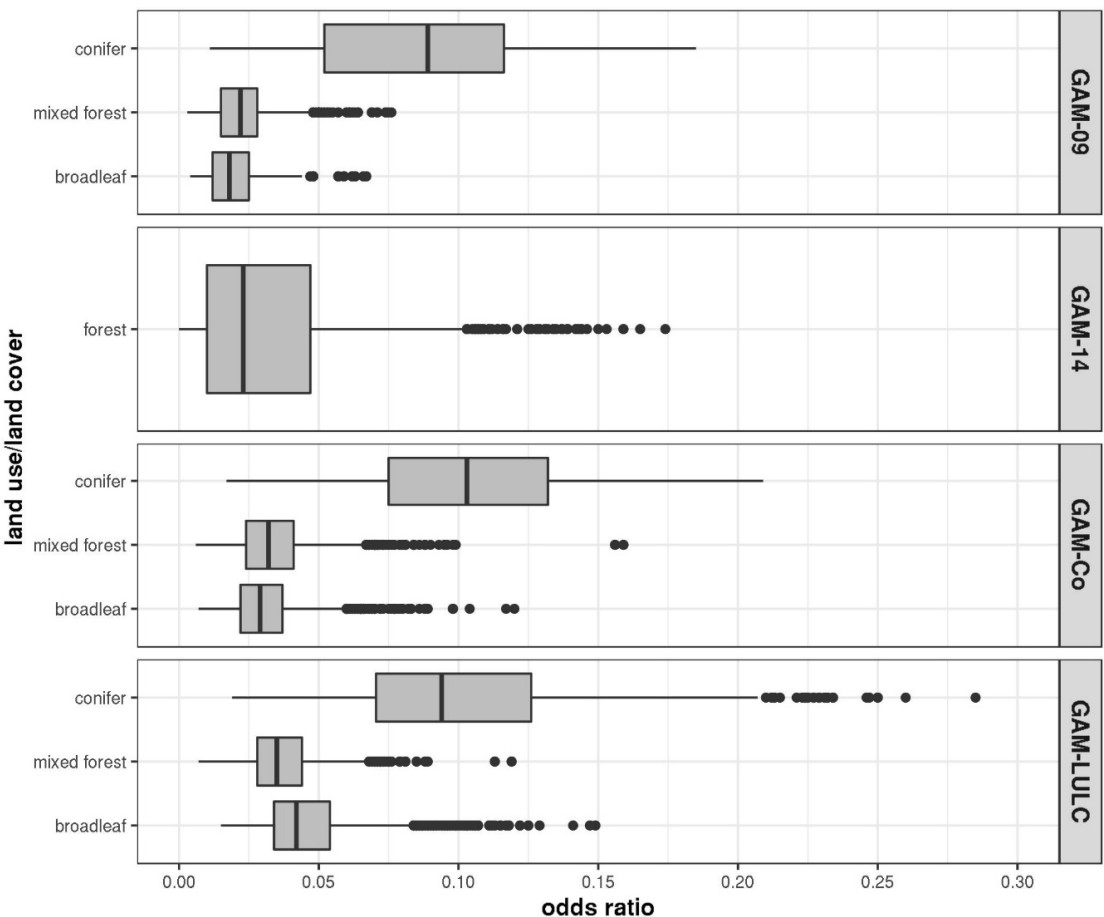

**Figure 7.** Relationship between landslide occurrence and LULC predictor forest type in the SpCV-models as estimated by the odds ratio. Only significant relationships ($p$-value < 0.05) were selected. The reference level is 'no forest.' Note that an odds ratio of 1 represents 'no relationship' and an odds ratio smaller than 1 is associated with lower odds of outcome.

## 4. Discussion

### 4.1. Model Performance and the Role of Time-Varying Variables

The application of semi-parametric, additive models to event-based landslide data revealed substantial differences in performance estimates and substantial effects of time-varying predictor variables representing LULC and extreme rainfall conditions. The models performed extremely well in learning spatial patterns of landslide occurrence (0.81–0.94 mAUROC), supporting the proposed sampling design. However, they had more heterogeneous performances in transferring those patterns from one event to another as revealed by spatio-temporal cross-validation (mAUROC 0.72–0.95). Among the 'event-based models,' the model with combined data (2009 and 2014) performed best in spatial and temporal prediction. The use of the smaller 2014 event data (139 landslides) for training the model showed weaker performance estimates in temporal prediction. We therefore advocate the combined use of landslide data from multiple events to ensure better generalization.

Comparing different feature sets ('comparison models') revealed that the inclusion of time-varying meteorological and LULC input variables was beneficial (mAUROC of GAM-Co: 0.94) compared to only using LSV and geological predictors (mAUROC of GAM-Base: 0.81). This finding supports Gariano



and Guzzetti's [19] recommendation to consider time-varying predictors. Non-linear relationships between landslides and the input variables were present, which justified the selection of GAMs with their ability to automatically detect such relationships, especially in precipitation as a predictor exhibiting thresholding behavior (Figure A4 for GAM-Co).

Furthermore, the variable importance assessment supported the high importance of time-varying LULC and meteorological variables (ranked top five in each model). In the 'event-based models,' five-day rainfall was the most important variable, which is consistent with the findings by [36]. The high importance of meteorological variables in event-based landslide modeling clearly indicated the need to more systematically account for such temporally and spatially variable confounders in future studies. This is also a necessary prerequisite for the application of landslide susceptibility models to forecast susceptibility following severe weather events and to predict susceptibility in future climate and LULC change scenarios.

The relationship between landslide occurrence and meteorological variables supports a critical rainfall threshold of around 50 mm for five-day rainfall, but reveals no distinctive threshold for maximum three-hour rainfall intensity, respectively. A comparison to other reported rainfall thresholds is challenging due to different thresholding approaches (e.g., based on maximum rainfall intensity or antecedent rainfall amount), the characteristics of rainfall events (e.g., spatial distribution), the type of landslides being studied and the geographic characteristics of the study areas (e.g., geology, topography, climate, vegetation) [24,88,89]. For the Austrian Alps, Moser and Hohensinn [90] discovered landslide occurrences with rainfall intensities from 20–100 mm/h for rainstorms of short duration to several hours, and of more than 2 mm/h for continuous heavy rainfall of one to two days (i.e., about 50–100 mm rainfall sum). Based on [90], Loibl et al. [91] explored a 40 mm/d exceedance for landslide occurrences in a Styrian municipality. We also compared our proposed threshold to others delineated for (pre-)Alpine regions [88,92], and found that it fits to the lower range of reported thresholds (e.g., 59.8 mm for Alpine mountain systems in Italy [92], or 63.1 mm for the Moscardo Torrent [93]; duration = 120 h). This comparable "conservative" threshold given the large amount of landslide occurrences, however, demonstrated the study area's high predisposition for landslides.

The relationship of landslide occurrences to the LULC variable forest type clearly indicated forest areas to be less landslide-prone than non-forest areas (e.g., OR = 0.01 for conifer forest in GAM-Co). Furthermore, we discovered significant differences between forest types as conifer forest was more prone to landslides than broadleaf or mixed forest. Similar findings were reported in [32]. However, the discussion of possible forest cover influences on slope stability is still ongoing [5,94]. These relationships may be scale-dependent (e.g., root system, species composition, tree weight [95]), but commonly, forest area is considered to mitigate landslide susceptibility [5,17,29]. More detailed forest cover data such as tree species, root depth, or tree weight as additional predictors may help to improve our understanding of its relationship to landslide occurrence as well as model fit.

### 4.2. Challenges and Uncertainties

In our analysis, we faced multiple challenges and discovered uncertainties regarding the sampling design, the data basis, and the extracted predictor-response relationships.

The spatiotemporal sampling of the landslide and non-landslide observations was subject to uncertainties. First, landslide inventories are never complete with an unknown level of incompleteness [8]. In particular, we believe that the inventory for the 2014 event is substantially incomplete, which may be one of the reasons for the high variability in model performance and predictor-response relationships. Moreover, since we generated the landslide-free mask based on the landslide observations, the effect of incompleteness could have been further propagated into the sampled absence locations (e.g., errors in reconstructing the effective surveyed area for the 2009 event). Second, the landslides recorded by IMG and LS were based on damage claims, which may have led to a potential inventory bias towards economically "valuable" objects while under-reporting landslides in remote areas, as also found for a different study area by Steger et al. [96]. We tried to reduce this

potential bias by considering a hypothetically driven road network (for 2009) and the creation of an affected area mask (for 2014), respectively. Considering these uncertainties, we emphasize the need to delineate the inventoried area at the time of data acquisition, e.g., by recording the surveyed route (and viewing distance) or ideally digitizing the inventoried area.

With the available data, we could confirm the relevance of commonly reported predictor variables in landslide susceptibility modeling [14,53,54,70]. However, several potentially important predictors are still missing. Whereas short and heavy rainfall events mainly trigger shallow landslides [97], long-lasting rainfall events trigger both shallow and deep-seated landslides [3,88,98]. A separate treatment of shallow and deep-seated landslides (e.g., by separate model frameworks or an additional predictor) may allow to better understand the effects of different types of rainfall events. However, the landslide type was not available for all observations in our study (Table A2). Furthermore, for soil reinforcement and slope stability, the local soil water status, regolith information, and plant characteristics play a major role [95,99]. From a climatological perspective, the consideration of only precipitation data for landslide susceptibility modeling may not be sufficient in a changing climate. Increased temperatures enhance evapotranspiration, which in turn may reduce landslide activity [3,100]. The inclusion of predictors such as soil material, soil moisture or evapotranspiration may further improve the understanding of the physical processes controlling slope stability. However, meteorological, pedological, or vegetation data are often not known at the local to regional scale and in sufficient detail. The available data resolution (INCA at 1 km × 1 km, HRDTM) allowed the applicability of the proposed approach at local scale (10 m × 10 m) [25,101]. On the slope scale, where geotechnical information or hydrological conditions, e.g., for monitoring, are available, physically based numerical models may be more appropriate [25,101].

As meteorological variables, we used cumulative event rainfall (five-day rainfall) and rainfall intensity at the failure date (maximum three-hour rainfall intensity). However, there are many possibilities how to aggregate meteorological data [88], ranging from indices (e.g., landslide-rainfall index [35], antecedent rainfall index [102]) to fixed moving sizes or number of days exceeding a certain amount of rainfall (e.g., in [34,36]), and there is no general agreement on an optimal aggregation scheme. However, the medium to high agreement between the aggregated meteorological variables and the landslide rain events according to [22], confirmed their applicability (correlation of 0.95 for five-day rainfall and of 0.58 for maximum three-hour rainfall intensity, respectively).

In the relationship between landslide occurrence and meteorological variables, we discovered a higher uncertainty for higher rainfall amount and intensity (>165 mm and 10 mm/h, respectively). We assume that the sampling design together with physical causes contributed to the uncertainties: Around 6.3% of the landslides occurred at the last day of the rainfall events (Table A3). With the proposed sampling approach, absence observations accounted for about 11.1% of. Using the sum of five-day rainfall, it may be that too much rain was attributed to these absence observations, leading to a bias for higher rainfall (Figure A2). Moreover, it is also possible that all landslides have already occurred for a certain rainfall, so that physically no further landslides can be expected. Again, absence observations beyond that certain rainfall may blur the underlying relationship. With more extreme rainfall-induced landslides available in future event inventories, the established relationships are expected to become more robust and thus reduce model uncertainties.

## 5. Conclusions and Outlook

The relationship of intense precipitation, LULC and landslide occurrence is still not fully understood. In this study, we developed a model and sampling design to analyze landslide occurrences after extreme rainfall events in 2009 and 2014. Our analysis focused on the relationship of slope failures to time-varying predictor variables (land use/land cover and meteorological variables) in statistical landslide susceptibility modeling using semi-parametric generalized additive models.

We successfully modeled rainfall-triggered landslides in space and time, but discovered limitations in model transferability in time when the event data was sparse. We conclude that the combined

modeling of multiple events was preferable to individual ones. Furthermore, we found a significant increase in model performance when time-varying predictor variables had been considered as model input. The relevance of time-varying predictors was also supported by the variable importance. The variables forest type, five-day rainfall, and maximum three-hour rainfall intensity were among the top five most important variables in explaining landslide occurrence. Regarding land use/land cover, we identified a lower chance for landslide occurrences in forest areas than outside forests, while accounting for the other variables in the model. Additionally, broadleaf or mixed forests were less prone to landslides than conifer forests. For the meteorological variables, both showed a different relationship to landslide occurrences: For the maximum three-hour intensity the general trend showed that the chance for landslide occurrences was higher with higher values; for the five-day rainfall, there was an increasing chance of landslide occurrence up to around 50 mm, where a plateau was identifiable. We related these relationships to critical rainfall amounts, suggesting that the threshold of five-day rainfall amount in the Styrian Basin is of the order of 50 mm.

A changing environment may alter future occurrence patterns and frequencies of landslides. Our establishment of a landslide susceptibility model and sampling design incorporating time-varying variables is a first step towards assessing time-varying landslide susceptibility. This means that the time-varying predictors can be easily exchanged with climate and LULC change scenario data, and thus storylines of future susceptibility can be calculated. This in turn enables the assessment of risk perceptions and adaptation intentions of the population in endangered areas. Additionally, it may provide new insights for effective risk management and communication to promote landslide preparedness on the individual and community levels (e.g., for drainage of slopes, emergency plans, and relocation).

Further research may focus on the validation and application of the identified threshold in landslide hazard or risk modeling, may explore model sensitivity towards different delineation strategies of the landslide-free mask, or may enhance the model's feature space (e.g., predictors for soil moisture, tree weight, or evapotranspiration).

**Author Contributions:** Conceptualization, R.K. and A.B.; Formal analysis, R.K.; Funding acquisition, D.M.; Methodology, R.K.; Project administration, D.M. and A.B.; Resources, H.P. (Herwig Proske), D.M. and A.B.; Software, R.K.; Supervision, H.P. (Helene Petschko), H.P. (Herwig Proske), and A.B.; Validation, R.K.; Visualization, R.K.; Writing—original draft, R.K.; Writing—review & editing, R.K., H.P. (Helene Petschko), H.P. (Herwig Proske), P.L., D.M. and A.B. All authors have read and agreed to the published version of the manuscript.

**Funding:** This research was conducted within the Eastern Alpine Slope Instabilities under Climate Change (EASICLIM) project funded by the Austrian Climate Research Program (ACRP), grant number KR16AC0K13160. We acknowledge support by the German Research Foundation and the Open Access Publication Fund of the Thueringer Universitaets- und Landesbibliothek Jena Projekt-Nr. 433052568.

**Acknowledgments:** We are grateful to the Federal State of Styria for providing the high-resolution digital elevation model, the watershed map, the geological base map and the 2014 landslide data. Furthermore, we thank the Geological Survey of Austria and the Institute of Military Geoinformation for providing the 2009 landslide data. We would also like to thank the Central Institute for Meteorology and Geodynamics for providing the INCA precipitation data. Furthermore, we would like to thank three anonymous reviewers for their valuable comments.

**Conflicts of Interest:** The authors declare no conflict of interest.

## Appendix A. Descriptive Summary of Input Data

**Table A1.** The study's database.

| Name | Description | Source |
|---|---|---|
| | *Climate Data* | |
| INCA | Integrated Nowcasting through Comprehensive Analysis | Central Institute for Meteorology and Geodynamics (ZAMG) |
| | *Landslide Data* | |
| Event 2009 | rainfall-triggered, 22 to 25 June | Geological Survey of Austria (GBA), Institute of Military Geoinformation (IMG) |
| Event 2014 | rainfall-triggered, 12 to 15 September | Department of Hydrology, Resources and Sustainability of the Styrian Government (LS) |
| | *Land Surface Data* | |
| HRDTM | airborne LiDAR-derived high resolution digital terrain model | GIS department of Styria (GIS-Steiermark) |
| LULC | forest map with conifer amount; classified in: no forest, broadleaf, mixed forest and conifer | JOANNEUM RESEARCH (JR) |
| Watershed | 1303 catchment areas | GIS department of Styria (GIS-Steiermark) |
| Geological base map | reference scale: 1:50,000; classified in geological units | GIS department of Styria (GIS-Steiermark), alluvial corrected by Austrian Institute of Technology (AIT) |
| OSM | OpenStreetMap data for Styria as of 29.10.2018 | Geofabrik, OpenStreetMap (OSM) |

**Table A2.** Landslide data.

| Description | Total | Event 2009 | | Event 2014 |
|---|---|---|---|---|
| | | GBA | IMG | LS |
| *Landslides* | | | | |
| presence (raw) | 626 | 218 (680) | 269 (1026) | 139 (508) |
| absence | 3130 | 2435 | | 695 |
| *Landslide Type (%)* | | | | |
| shallow | 421 (67.3) | 414 (85) | | 7 (5) |
| deep-seated | 157 (25.1) | 73 (15) | | 84 (60.4) |
| Failure date (%) | 245 (39.1) | 106 (21.8) | | 139 (100) |
| *LULC (%)* | | | | |
| forest | 78 (12.5) | 68 (14) | | 10 (7.2) |
| OSM | | | | vineyard and orchard (30.8%), farmland (22.2%), urban areas (streets and buildings, 30.3%), meadow and forests (16.8%) |

**Table A3.** Temporal distribution of landslide presence and absence observations.

| Description | Dates | | | | | | | | | | | | | | | | |
|---|---|---|---|---|---|---|---|---|---|---|---|---|---|---|---|---|---|
| | *A: Event 2009: 16–26 June* | | | | | | | | | | | | | | | | |
| *D* | 19 | 20 | 21 | 22 | 23 | 24 | 25 | 26 | | | | | | | | | |
| n | 304 | 313 | 302 | 283 | 325 | 542 | 511 | 342 | | | | | | | | | |
| $P_o$ | 0 | 0 | 0 | 0 | 8 | 36 | 36 | 4 | | | | | | | | | |
| P | 0 | 0 | 0 | 0 | 43 | 225 | 200 | 19 | | | | | | | | | |
| A | 304 | 313 | 302 | 283 | 282 | 317 | 311 | 323 | | | | | | | | | |
| | *B: Event 2014: 30 August–15 September* | | | | | | | | | | | | | | | | |
| *D* | 30 | 31 | 01 | 02 | 03 | 04 | 05 | 06 | 07 | 08 | 09 | 10 | 11 | 12 | 13 | 14 | 15 |
| n | 4 | 6 | 5 | 9 | 13 | 7 | 7 | 6 | 6 | 3 | 116 | 123 | 96 | 87 | 186 | 125 | 35 |
| P, $P_o$ | 0 | 0 | 0 | 0 | 0 | 0 | 0 | 0 | 0 | 0 | 0 | 0 | 0 | 6 | 99 | 24 | 10 |
| A | 4 | 6 | 5 | 9 | 13 | 7 | 7 | 6 | 6 | 3 | 116 | 123 | 96 | 81 | 87 | 101 | 25 |

**D**: Dates, **n**: number of observations, $P_o$: original distribution of failure dates (GBA, IMG, and LS records), **P**: landslide presence, **A**: landslide absence.

**Table A4.** Summary of predictor variables for combined-event data.

| Variable | n (%) | Min | $q_1$ | $\tilde{x}$ | $\bar{x}$ | $q_3$ | Max | IQR |
|---|---|---|---|---|---|---|---|---|
| *land surface variables* | | | | | | | | |
| convergence index, 100 m | 3756 | −54.19 | −12.05 | 3.41 | 3.97 | 19.65 | 71.45 | 31.71 |
| convergence index, 500 m | 3756 | −46.46 | −8.84 | 4.1 | 3.91 | 16.89 | 58.79 | 25.73 |
| curvature, plan | 3756 | −0.086 | −0.002 | 0 | 0 | 0.003 | 0.06 | 0.006 |
| curvature, profile | 3756 | −0.072 | −0.003 | 0.001 | 0.001 | 0.005 | 0.051 | 0.007 |
| flow accumulation | 3756 | 4.61 | 5.96 | 6.53 | 6.67 | 7.16 | 16.73 | 1.2 |
| normalized height | 3756 | 0.04 | 0.24 | 0.48 | 0.49 | 0.72 | 0.96 | 0.48 |
| slope angle | 3756 | 0.09 | 7.36 | 12.01 | 12.77 | 17.2 | 42.32 | 9.84 |
| slope angle, catchment | 3756 | 0.2 | 8.83 | 11.87 | 11.82 | 14.48 | 33.53 | 5.65 |
| slope aspect, S-N | 3756 | −1 | −0.69 | −0.11 | −0.06 | 0.54 | 1 | 1.23 |
| slope aspect, W-E | 3756 | −1 | −0.77 | 0.1 | 0.04 | 0.8 | 1 | 1.57 |
| TPI | 3756 | −12.67 | −0.73 | 0.08 | 0.17 | 1.11 | 8.25 | 1.84 |
| SWI | 3756 | 1.21 | 2.92 | 3.29 | 3.44 | 3.8 | 7.98 | 0.89 |
| *meteorological variables* | | | | | | | | |
| five-day rainfall | 3756 | 3.46 | 27.32 | 52.28 | 76.77 | 131.06 | 182.18 | 103.74 |
| maximum three-hour rainfall intensity | 3756 | 0 | 0.51 | 1.93 | 3.33 | 4.07 | 17.51 | 3.57 |
| *geology* | | | | | | | | |
| 0 | 575 (15.3) | | | | | | | |
| 1 | 1108 (29.5) | | | | | | | |
| 2 | 1753 (46.7) | | | | | | | |
| 3 | 320 (8.5) | | | | | | | |
| *land use/land cover forest type* | | | | | | | | |
| no forest | 2527 (68.3) | | | | | | | |
| broadleaf | 503 (13.4) | | | | | | | |
| mixed forest | 590 (15.7) | | | | | | | |
| conifer | 136 (3.6) | | | | | | | |

Statistic shows observation numbers (**n**), minimum (**Min**), 25th percentile (**$q_1$**), median ($\tilde{x}$), mean ($\bar{x}$), 75th percentile (**$q_3$**), maximum (**Max**), interquartile range (**IQR**); For nominal predictor variables the percentage of observation per class is given (%); Geological units: 0: other units, 1: Neogene formations dominated by fine-grained sediments, 2: Neogene formations with coarse-grained layers, and 3: pre-Würm pleistocene formations.

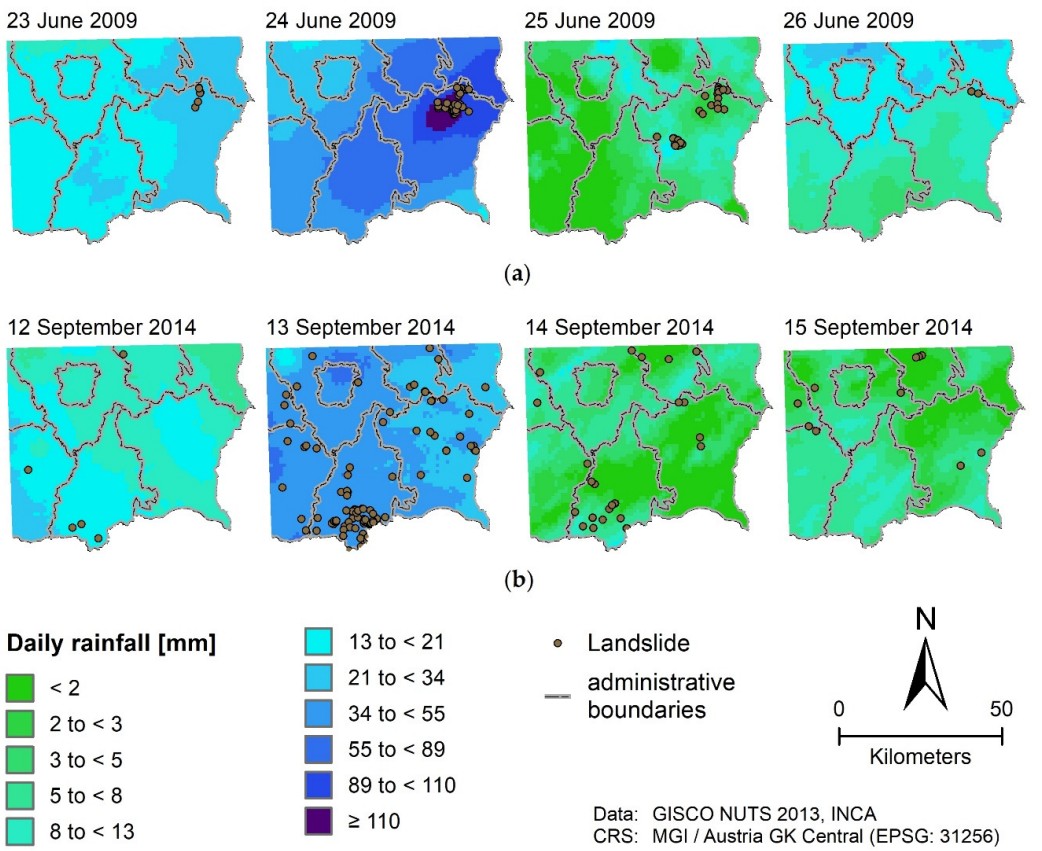

**Figure A1.** Total daily rainfall [mm] of days with landslide occurrences: (**a**) 2009 event; (**b**) 2014 event.

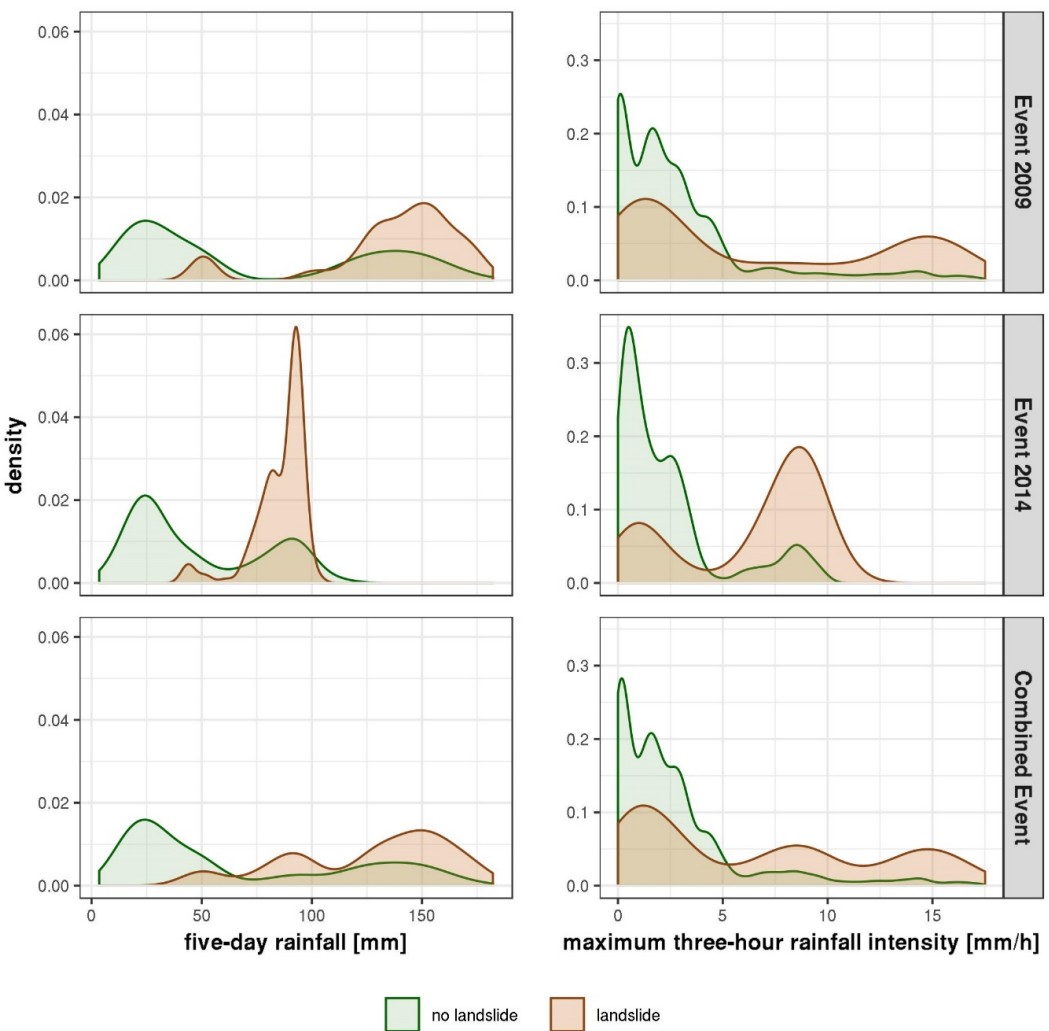

**Figure A2.** Density plots showing the distribution of meteorological variables dependent on the event data, grouped by landslide and non-landslide occurrences.

## Appendix B. Descriptive Summary of Processing Results

**Table A5.** Folds-based model performances.

| Model | $\tilde{x}$ | $\bar{x}$ | Range | Min | Max | IQR |
|---|---|---|---|---|---|---|
| *A: Spatial-Cross Validation (SpCV)* | | | | | | |
| GAM-09 [1] | 0.94 | 0.94 | 0.14 | 0.84 | 0.98 | 0.04 |
| GAM-14 [1] | 0.92 | 0.90 | 0.56 | 0.44 | 1.00 | 0.07 |
| GAM-Co [1,2] | 0.94 | 0.94 | 0.15 | 0.84 | 0.99 | 0.02 |
| *B: Spatio-Temporal Cross Validation (SpTempCV)* | | | | | | |
| GAM-09 predicting 2014 | 0.89 | 0.89 | 0.08 | 0.84 | 0.92 | 0.01 |
| GAM-14 predicting 2009 | 0.72 | 0.69 | 0.66 | 0.28 | 0.94 | 0.31 |
| GAM-Co predicting 2009 | 0.95 | 0.94 | 0.08 | 0.89 | 0.97 | 0.02 |
| GAM-Co predicting 2014 | 0.91 | 0.91 | 0.15 | 0.83 | 0.98 | 0.03 |
| *C: Spatial-Cross Validation Comparison (SpCV-Comparison)* | | | | | | |
| GAM-Base [2] | 0.81 | 0.80 | 0.30 | 0.64 | 0.94 | 0.06 |
| GAM-LULC [2] | 0.87 | 0.86 | 0.28 | 0.67 | 0.95 | 0.06 |
| GAM-Meteo [2] | 0.91 | 0.91 | 0.19 | 0.80 | 1.00 | 0.03 |
| GAM-Co [1,2] | 0.94 | 0.94 | 0.15 | 0.84 | 0.99 | 0.02 |

[1] event-based models; [2] comparison-models; Statistic: minimum (**Min**), median ($\tilde{x}$), ($\bar{x}$), range (Range), maximum (**Max**), interquartile range (**IQR**).

**Table A6.** Wilcoxon signed rank tests for SpCV-Comparison.

| Model | Z | *p* -Values | Median (Δ) | *r* |
|---|---|---|---|---|
| GAM-Base | | | 0.81 | |
| GAM-LULC | 40.91 | <0.001 | 0.87 (0.06) | 0.82 |
| GAM-Meteo | 42.58 | <0.001 | 0.91 (0.04) | 0.85 |
| GAM-Co | 42.48 | <0.001 | 0.94 (0.03) | 0.85 |

**Z**: Z score; Number of observations per group: 2500, alternative hypothesis: *greater*, $\alpha = 0.05$.

**Table A7.** Variable importance.

| Variable | GAM-09 | GAM-14 | GAM-Co | GAM-LULC | GAM-Meteo | GAM-Base |
|---|---|---|---|---|---|---|
| *land surface variables* | | | | | | |
| convergence index, 100 m | 0.39 (12) | 1.37 (10) | 0.34 (10) | 0.46 (8) | 0.34 (11) | 0.39 (10) |
| convergence index, 500 m | 0.43 (10) | 1.46 (8) | 0.3 (11) | 0.15 (12) | 0.28 (12) | 0.14 (13) |
| curvature, plan | 0.25 (14) | 1.23 (12) | 0.21 (14) | 0.4 (9) | 0.24 (14) | 0.31 (11) |
| curvature, profile | 1.67 (4) | 1.43 (9) | 1.21 (4) | 1.41 (3) | 0.92 (4) | 1.35 (2) |
| flow accumulation | 0.39 (13) | 0.94 (14) | 0.27 (13) | 0.32 (11) | 0.28 (13) | 0.54 (9) |
| normalized height | 0.42 (11) | 2.12 (6) | 0.55 (9) | 1.4 (4) | 0.59 (6) | 1.33 (3) |
| slope angle | 7.05 (3) | 2.03 (7) | 5.31 (3) | 7.85 (2) | 3.87 (2) | 5.96 (1) |
| slope angle, catchment area | 0.17 (16) | 0.64 (16) | 0.08 (16) | 0.12 (13) | 0.11 (15) | 0.2 (12) |
| slope aspect, S-N | 0.17 (15) | 1.35 (11) | 0.09 (15) | 0.1 (14) | 0.75 (5) | 1.06 (4) |
| slope aspect, W-E | 0.92 (6) | 0.81 (15) | 0.27 (12) | 0.35 (10) | 0.38 (9) | 0.56 (8) |
| TPI | 0.54 (9) | 2.37 (4) | 0.66 (8) | 1.06 (6) | 0.37 (10) | 0.69 (6) |
| SWI | 0.74 (7) | 0.95 (13) | 0.71 (7) | 0.88 (7) | 0.5 (8) | 0.59 (7) |
| *meteorological variables* | | | | | | |
| five-day rainfall | 13.57 (1) | 17.75 (1) | 15.13 (1) | | 16.85 (1) | |
| maximum three-hour rainfall intensity | 1.58 (5) | 2.91 (3) | 1.05 (5) | | 1.75 (3) | |
| *geology* | | | | | | |
| geology | 0.55 (8) | 2.27 (5) | 0.73 (6) | 1.18 (5) | 0.57 (7) | 0.77 (5) |
| *land use/land cover* | | | | | | |
| forest type | 9.31 (2) | 4.14 (2) | 8.57 (2) | 12.45 (1) | | |

Variable importance measured in in mean decrease in deviance explained (%), rank of variable in parentheses.

**Table A8.** Wilcoxon signed rank test for OR of LULC variable forest type.

| Class | Z | N | *p* -Values | Median* (Δ) | OR | *r* |
|---|---|---|---|---|---|---|
| | | | *A: GAM-09* | | | |
| conifer | | | | −2.42 | | |
| mixed forest | −33.31 | 1484 | <0.001 | −3.86 (−1.44) | 0.24 | −0.86 |
| broadleaf | −14.42 | 2500 | <0.001 | −4.07 (−0.21) | 0.81 | −0.29 |
| | | | *B: GAM-Co* | | | |
| conifer | | | | −2.27 | | |
| mixed forest | −33.29 | 1608 | <0.001 | −3.44 (−1.17) | 0.31 | −0.83 |
| broadleaf | −7.33 | 2500 | <0.001 | −3.61 (−0.17) | 0.84 | −0.14 |
| | | | *C: GAM-LULC* | | | |
| conifer | | | | −2.36 | | |
| broadleaf | −41.23 | 2375 | <0.001 | −3.19 (−0.83) | 0.44 | −0.85 |
| mixed forest | −20.15 | 2500 | <0.001 | −3.32 (−0.14) | 0.87 | −0.4 |

**\***: Log-transformed OR (i.e., logit scale); **N**: Number of observations per group; **Z**: Z score; Alternative hypothesis: *less*, $\alpha = 0.05$.

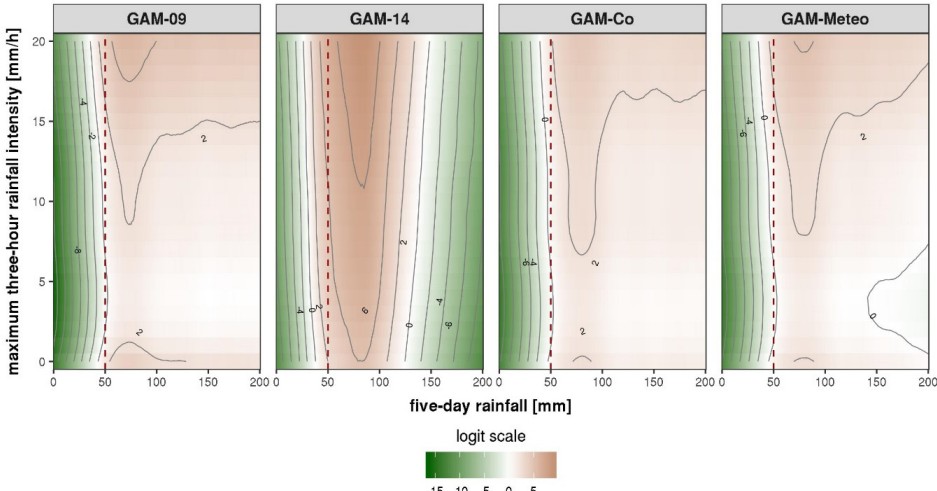

**Figure A3.** Relationship between landslide occurrence and metrological variables as pair of predictor variables. The relationship on the linear predictor scale (logit scale, centered) were extracted from all SpCV models and averaged (contour lines). Red dashed line: Estimated critical rainfall thresholds for five-day rainfall.

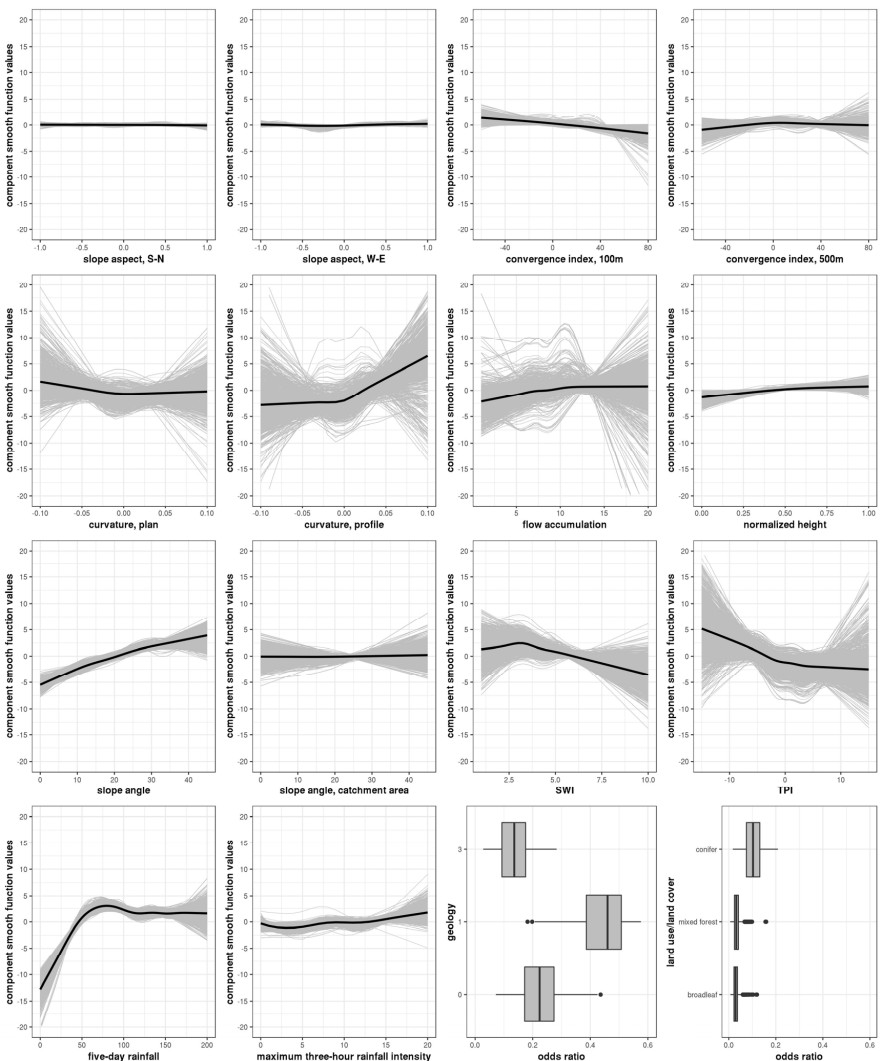

**Figure A4.** Component smooth function plots for GAM-Co; Geological units: Reference: Neogene

formations with coarse-grained layers, 0: other units, 1: Neogene formations dominated by fine-grained sediments, and 3: pre-Würmian Pleistocene formations; LULC classes: Reference: no forest.

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
