# Peer review of "Event-Based Landslide Modeling in the Styrian Basin, Austria: Accounting for Time-Varying Rainfall and Land Cover"

_geosciences, doi:10.3390/geosciences10060217_

Round 1
Reviewer 1 Report
Starting from data of diffused landslides caused by the two major rainstorm events of 2009 and 2014, the manuscript deals with a very interesting topic which is focused on the analysis of landslide susceptibility of the Styrian Basin (Austria) by advanced statistical approaches. The approaches proposed are novel in the field of landslide science, especially for having integrated in Generalized Additive Models (GAM) spatial and temporal variability of a series of factors controlling landslide phenomena such as geology, meteorology, land cover and landuse (LUCL variables) as well as for having considered also features of areas in with no landslide occurrences. Results of the modelling are of high quality as demonstrated by very high value of AUROC tests.
In the opinion of this reviewer the manuscript is scientifically sound, well balanced and well written.
Few improvements can be suggested by this Reviewer such as indicated following in terms of general and specific comments.
General comments
- The integration of the description of rainstorm events with isohyet map and description of maximum intensities recorded will help understanding the severity of 2009 and 2014 meteorological events as well as their spatial distribution.
- A more detailed description of landslide types, should be carried out in par. 2.2.3, in terms of classification, depth and materials involved, even with a specific new figure based on comprehensive descriptive statistics which could improve the comprehension of the impact of the two extreme meteorological events.
- A more detailed description of rainfall thresholds (Intensity vs Duration or Cumulated rainfall vs Duration), eventually established for the study area by other studies, will help to understand the severity of 2009 and 2014 rainfall events, not excluding a graphical comparison.
- The integration of discussion with expected problems and advantages related to the application of the approach proposed at different scales, especially at more detailed ones, will help the reader to understand limits and possibility of applicability.
- To integrate discussion with a more consistent analysis of results showed in Fig. B1 in which the landslide occurrence is showed as concentrated around fixed values of rainfall (from a specific maximum value to different local maxima) instead of showing a progressive increase as the rainfall growths.
Specific comments
Line 100: to correct “chose” with “choose”
Line 113: is the unit used for relief energy relief correct?
Reviewer 2 Report
The manuscript contains many details and informations on landslide events of 2009 and 2014 occurred in Styria region, Austria. However, as one of the purpose of the manuscript is the temporal occurrence of landslides, I suggest to try to provide further details on results found; they seem confined in Table 3A and figure B1, which need further explanations to descrive the plot of figure B1. A description of rainfall distribution of storm events, over the most important climate station would improve the presentation.Author Response
Please see the attachment.

Reviewer 3 Report
The manuscript by Knevels et al. provides a landslide susceptibility analysis employing a spatio-temporal approach and focuses on meteorological and LULC data mainly. Their conclusions confirm earlier results wrt. the connection between meteorological variables and slope failure. The novelty of this work is in the methodological approach of spatio-temporal sampling (and the related performance analysis).
The presentation quality of this work is high, consisting of a concise and valuable exposition, an outstanding methods section as well as well-written descriptions and well-prepared illustrations. The discussion and conclusion sections fall a bit behind in comparison, and I would have a few comments regarding the discussion and presentation of results.
L18 + 548: "The interplay...". This statement feels a bit arbitrary and it would benefit from being a bit more specific.
L159: The statement on the correction procedure feels a bit ambiguous. In particular as any publishing information about this map is missing. If it was not published at all, what is the year of creation, so that the extent of modifications become a bit clearer.
L166: "To account for the approximate spatial extent of linear objects". Why is OSM data considered approximate?
L271: 140 m does not seem arbitrary at all. Can you explain? Apparently spatial effects of auto-correlation was your criterion to set the minimum distance?
The discussion section contains quite a bit of outlook material, while the outlook section lacks this. In particular if you look at L493, L519, L526, L555 -- and I might have missed other occurrences -- target at future tasks. These are, however, not detailed any further, and thus they remain statements without revealing how this could be achieved. E.g. L 536: "it would be highly desirable...". This triggers some questions: why hasn't that data been included here, where could that data come from in an improved version, what are the actual benefits and what effects would you expect. Or L545: what do you mean by "further investigations" exactly. I suggest to revisit these statements and add some more specific information.
I am not saying these statements are incorrect, but I would strongly suggest to discuss them actually, as this is a discussion section, and move the more general emphasis of their importance into the conclusion's outlook section, that is currently lacking these kind of "visions".
L569: you write this to open the door for your statement on L570, yet the connection is not really clear to me. Can you add some concrete statements, in particular to the importance of temporal assessments?
Also the statement on L573: "new insights for effective risk management" could be a bit better specified as it currently reads quite generic.
